# Mechanical Properties of Carbon Fiber-Reinforced Polymer Concrete with Different Polymer–Cement Ratios

**DOI:** 10.3390/ma12213530

**Published:** 2019-10-28

**Authors:** Gao-Jie Liu, Er-Lei Bai, Jin-Yu Xu, Ning Yang

**Affiliations:** 1Department of Airfield and Building Engineering, Air Force Engineering University, Xi’an 710038, China; liusiremail@163.com (G.-J.L.); eihuiyi@163.com (J.-Y.X.); 17749132826@163.com (N.Y.); 2College of Mechanics and Civil Architecture, Northwest Polytechnic University, Xi’an 710072, China

**Keywords:** emulsion powder, carbon fiber-reinforced polymer concrete, mechanical properties, polymer–cement ratio, toughening and crack resistance

## Abstract

To study the effect of redispersible polymer emulsion powder on the mechanical properties of carbon fiber-reinforced polymer concrete (CFRPC), the compressive, flexural, and splitting tests of CFRPC specimens with different polymer–cement ratios (polymer–cement mass ratios) were performed in this study. The modification effect of emulsion powder on CFRPC was analyzed from the perspectives of the strength and deformation properties of the specimens. The results show that the static properties of CFRPC increased first and then decreased with the increase of the polymer–cement ratio, in which the splitting tensile strength had the most significant increase; the flexural strength took second place and the compressive strength had a slight increase. When the polymer–cement ratio was 8%, the flexural and splitting tensile strength of the CFRPC specimens increased significantly by 36% and 61%, respectively. According to electron microscopy images, adding emulsion powder can effectively improve the structure of fiber–matrix transition zones and enhance the bond property between fibers and the matrix.

## 1. Introduction

Due to the good strength and constructability, concrete is widely used in construction engineering as an important building material. However, ordinary concrete still has some limitations, such as poor crack resistance and low tensile and flexural strength [1]. Considering the advantages of high strength, high elastic modulus, and corrosion resistance, carbon fibers are usually used to strengthen and toughen brittle materials [2]. Based on this, researchers modified the performance of concrete by adding carbon fiber. In carbon fiber-reinforced concrete (CFRC), the cohesion between carbon fiber and the cement paste matrix is mainly dependent on the surface energy of the materials. Combined with mechanical meshing, CFRC is strengthened and toughened to limit the crack generation [3,4]. However, due to the bond property between carbon fibers and the cement paste matrix, the carbon fibers can easily be pulled out or slip when the concrete is damaged. In this condition, the enhancement and reinforcement of crack resistance are not sufficient [5,6].

Emulsion powder is a type of powder binder with the advantages of high cohesive force, plasticity, and construction [7]. In recent years, the bond property of emulsion powder has been widely studied. Ohama and Beeldens [8,9,10] proposed an Ohama model for PCC (polymer cement concrete). The process of PCC formation was divided into three stages from the perspective of a microstructure. The chemical reaction and bond toughening of polymer in cement paste were discussed, and the model was continuously improved thereafter. Wang and Liu [11,12] analyzed the principle of the Ohama and B–O–V models. The secondary film formation and skin formation of water-soluble polymers such as emulsion powder at the internal and external interface of the cement concrete matrix were illustrated. In addition, the mechanical strength properties of styrene–acrylic copolymer emulsion powder cement mortar were studied. The results showed that the added emulsion powder can obviously improve the tensile and flexural properties of cement mortar. Taking steel fiber concrete as the matrix material, Hai et al. [13] considered the effect of polymer emulsion on the anti-penetration performance of concrete. The results showed that the emulsion powder was well dispersed at the transition interface after mixing with water, cement, and aggregate, and a polymer film with superior adhesion can be produced. The tensile strength, bending strength, deformation, and toughness of the concrete also increased.

At present, there are few studies on the composite modified concrete of carbon fiber and polymer at home and abroad. Based on the above problems, redispersible emulsion powder was added into concrete to effectively increase the cementation of cement paste matrix [13,14,15]. The micro-structure of the interior interface transition zone of concrete was improved [16,17,18], and the bonding force between carbon fiber and the cement paste matrix was enhanced [19]. Therefore, it is of great significance to study the modification effect of emulsion powder on CFRC.

Based on this, static mechanical tests of CFRPC specimens with 0.1% carbon fiber content (volume ratio of carbon fiber to concrete) were conducted with a electro-hydraulic servo material test system. By testing the compressive, flexural, splitting tensile strength, and peak strain of specimens, different polymer–cement ratios were studied from the perspective of strength and deformation properties. The influence of the polymer–cement ratio on the compressive, splitting, and flexural properties of CFRPC was studied. The modification mechanism of emulsion powder was discussed based on the observation of the micro-morphology. In addition, concrete specimens with 0.2% carbon fiber content were performed as supplementary experiments. The reliability of the test results was verified by comparing the relationship between the static mechanical properties and polymer–cement ratio of concrete with two kinds of carbon fiber contents.

## 2. Materials and Methods 

### 2.1. Test Materials and Mixing Ratio

The specific technical route is shown in Figure 1.

A 42.5 grade ordinary Portland cement was used in the test. Running water was used to mix the cement, and the water quality met the testing standard. Well-graded limestone rubble less than 20 mm was used as coarse aggregate, and the bulk density was 2700 kg/m^3^, bulk density was 1600 kg/m^3^, and mud content was 0.2%. Ordinary river sand with a fineness modulus of 2.8, a bulk density of 1503 kg/m^3^, and a mud content of 1.5% was taken as the fine aggregate. 

Concrete additives (<5% of the cement mass) were added in the process of mixing concrete to improve its performance. Specifically, a water-reducing agent is a kind of concrete additive that can reduce the water consumption of concrete while keeping the slump of concrete basically unchanged. As a brown powder, a FDN superplasticizer water-reducing agent mother liquor was used, which was a multi-naphthalene nuclear sulfonated sodium sulphate with sulfuric acid after naphthalene sulfonation, with condensation by formaldehyde afterwards. Defoamer is a kind of concrete additive that can reduce surface tension and restrain or eliminate froth. Metal soap defoamer was used, which was composed of aluminum stearate, calcium stearate, potassium oleate, and calcium oleate. A film-forming additive can promote plastic flow and elastic deformation of polymer compounds and, thus, improve the coalescence properties. The DN-12 film-forming additive (2,2,4-trimethyl-1,3-pentanediol monoisobutyrate; C_12_H_24_O_3_) was used. Dispersant can reduce carbon fiber aggregation during concrete mixing. Hydroxyethyl cellulose (C_2_H_6_O_2_·x) dispersant was used, which is a white or yellowish, tasteless, non-toxic, fibrous or powdered solid, prepared by etherification of basic cellulose and ethylene oxide (or chloroethanol).

The VINNAPAS®5044N redispersible emulsion powder produced by the Wacker Company of Germany was used. Table 1 and Figure 2 show the main performance indexes and appearances, respectively. Table 2 and Figure 3 show the main properties and the appearance of PAN (polyacrylonitrile)-based short-cut carbon fibers, respectively. These data were obtained from the manufacturer.

To ensure the reliability of the test results, CF01RC and CF02RC carbon fiber concretes were designed as comparative tests. Table 3 shows the mix designs of two groups of concrete.

The dosage of coalescing agents was obtained by the mass ratio of film-forming additives to polymer mass. Based on experimental results, when the dosage of film-forming additives was 5%, the film-forming additives can significantly improve the polymer film-forming situation. In this study, the dosage of fixed film-forming additives was 5%, which was used as the research variable together with the dosage of emulsion powder. In addition, the dosage ratio of film-forming additives to the total mass of concrete was less than 10^-3^ orders of magnitude. Therefore, the dosage of film-forming additives can be ignored as a single variable for the mechanical properties of concrete.

### 2.2. Sample Preparation and Test Scheme

The carbon fiber dispersion was prepared. Firstly, water was poured into the blender and stirred at low speed (120 r/min), then dispersant was added with a slow stir. When the solution became a uniform gel, the solution was cooled to room temperature without stirring. Then, chopped carbon fibers were slowly added into the dispersant solution, stirring continued for approximately 10 min. After that, a water reducing agent, defoamer, and film-forming additive were added; the solution was then stirred at low speed for 60 s. Finally, the carbon fiber dispersion with good dispersion was prepared.

Redispersible polymer emulsion powder was manually mixed with cement until the powder was evenly dispersed. Fine aggregate and half of the carbon fiber dispersion liquid were poured into a concrete blender and stirred for 60 s. Coarse aggregate was added and stirred for 60 s; dispersed powder was added to the blender and stirred for 60 s; then, the remaining carbon fiber dispersion liquid was added and stirred for 60 s. The weighed defoamer was poured into the blender and stirred for 120 s. Then, the concrete was poured out of the mixer and manually poured for 60 s until the required slump was obtained. The standard curing was performed for 28 days after the model was loaded and vibrated.

The static test of concrete was carried out using the HYY electro-hydraulic servo material test system. Three test methods were used, including compressive test, splitting test and flexural test of concrete. The specimen was a cube with a side length of 100 mm. Nova NanoSEM230 field emission scanning electron microscopy (produced by the American FEI Company) was used to observe the micro-morphology of specimen slices. 

Mechanical test details are as follows [20]. The HYY electro-hydraulic servo material test system consists of a hydraulic test instrument and a data acquisition system. There are mainly three tests in the system, namely, comprehensive, splitting, and flexural tests.
Compressive test: the instrument was used to apply load uniformly on the cured specimens at a loading speed of 0.5–0.8 MPa/s until the specimens were damaged and the corresponding load was recorded;Splitting test: The preserved specimen, pad block, pad strip, and bracket were placed and installed in line with the requirements. Then, the hydraulic tester was used to apply load on the specimen uniformly at a loading speed of 0.05~0.08 MPa/s until the specimen failed; the corresponding failure load and displacement were recorded;Flexural test: the cured specimen, support, and hard steel cylinder were placed and installed in line with the requirements, referring to the method for the splitting test.

To reduce factors associated with accidental errors, three specimens were tested in each group.

To observe the specimen’s dimension under the scanning electron microscope, certain requirements should be observed. Accordingly, cured CFRPC specimens were cut from the middle into several cubes with side lengths of no more than 15 mm (the edge failure was inevitably encountered in the cutting process, which rarely affects the observation). Cubic specimens were selected according to the following requirements: (1) complete shape; (2) smooth surface; and (3) aggregate and the substrate can be observed. Then these samples were further polished by the sandpaper, after which they were cleaned and sealed in the bag for later use.

A vacuum (9 × 10^−3^ Pa) in the sample chamber was required for the normal operation of the scanning electron microscope. However, there are many internal pores in the CFRPC sample, which increases the difficulty of vacuuming. Accordingly, these specimens were dried in the oven for 6 hours at 50 °C. Disregarding the doped chopped carbon fiber in the CFRPC material, the overall conductivity was still poor in the dry state. Therefore, gold spraying on the observation surface of the specimen was implemented to facilitate observation under the scanning electron microscope. 

After the above operations, the CFRPC specimen could be observed clearly. Subsequently, the specimen was brought into the operation room and fixed on the sample stage with the gold-sprayed surface upward, then the height gauge was used to limit the height of the sample stand. The highest point of the sample and the lower edge of the gauge should be flush; and the sample table should be tight with the adjusting screw, fastening disc, and the base of the sample, followed by observation under the scanning electron microscope in the sample chamber.

## 3. Results

### 3.1. Compressive Property of CFRPC

Figure 4 shows the variation law of the influence of the polymer–cement ratio on the compressive strength of specimens. With the increase of the polymer–cement ratio, the compressive strength of the CFRPC01 specimens first increased and then decreased. When the polymer–cement ratio was 4%, the compressive strength of the specimens reached a peak value of 38.1 MPa, which was 10% higher than that of the specimens without emulsion powder. However, the strength of the specimens decreased by 8% as the content of the emulsion powder increased to 12%. Figure 5 shows the variation law of the peak strain of specimens with different polymer–cement ratios in the compressive tests. With the increase of the polymer–cement ratio, the peak strain of the CFRPC01 specimens increased continuously. When the polymer–cement ratio was 4%, 8%, and 12%, peak strain increased by 16%, 22%, and 38%, respectively. 

Compared with the CFRPC01 group, the change trends in compressive strength and peak strain of the CFRPC02 group were similar. When the emulsion powder was 4%, the strength reached its peak value, and the strength increased by 7%. With the increase in the polymer–cement ratio, the peak strain of the specimens continued to rise. It indicates that the polymer–cement ratio had little effect on the compressive strength of the specimens, but it had a significant impact on the compressive deformation performance.

### 3.2. Flexural Resistance of CFRPC

Figure 6 shows the influence of the polymer–cement ratio on the flexural strength of the specimens. With the increase in the polymer–cement ratio, the flexural strength of the CFRPC01 group first increased and then decreased. When the polymer–cement ratio was 8%, the flexural strength of the specimens reached a peak value of 8.6 MPa, which was 36% higher than that of specimens without emulsion powder. Figure 7 shows the variation law of the peak strain of the specimens affected by different polymer–cement ratios in flexural tests. With the increase of the polymer–cement ratio, the peak strain of the CFRC01 specimens first increased and then decreased. When the polymer–cement ratio was 4%, 8%, and 12%, peak strain increased by 27%, 45%, and 41%, respectively. Figure 8 shows the influence of the polymer–cement ratio on the flexural–compressive ratio of the specimens. With an increase in the polymer–cement ratio, the flexural–compressive ratio of the CFRPC01 group specimens increased constantly. When the polymer–cement ratio was 4%, 8%, and 12%, the flexural–compressive ratio increased by 6%, 34%, and 46%, respectively.

The flexural strength of the CFRPC01 and CFRPC02 specimens reached its peak value when the polymer–cement ratio was 8%, the peak strain first increased and then decreased, and the flexural–compressive ratio kept rising. It shows that the flexural strength of the CFRPC01 and CFRPC02 specimens increased with the increase in the polymer–cement ratio. In addition, with the increase in the polymer–cement ratio, the increase in the flexural strength was greater than that of the compressive strength.

### 3.3. Splitting Properties of CFRPC

Figure 9 shows the variation rule of the splitting tensile strength affected by the polymer–cement ratio. With the increase in the polymer–cement ratio, the splitting tensile strength of the CFRPC01 group first increased and then decreased. When the polymer–cement ratio was 8%, the splitting tensile strength of the specimens reached a peak value of 4.6 MPa, which was 62% higher than that of specimens without emulsion powder. Figure 10 shows the variation law of the peak strain of specimens affected by different polymer–cement ratios in splitting tensile tests. With the increase in the polymer–cement ratio, the peak strain of the CFRPC01 specimens increased slowly. When the polymer–cement ratio was 4%, 8%, and 12%, the peak strain increased by 1%, 4%, and 8%, respectively. Figure 11 shows the influence of the polymer–cement ratio on the tensile–compressive ratio of the specimens. With the increase in the polymer–cement ratio, the tensile–compressive ratio of the CFRPC01 group specimens increased constantly. When the polymer–cement ratio was 4%, 8%, and 12%, the tensile–compressive ratio increased by 6%, 34%, and 46%, respectively.

Compared with the splitting tensile test results of the CFRPC02 group specimens, it can be seen that the splitting tensile strength of the specimens was significantly affected by the addition of emulsion powder. When the polymer–cement ratio was 8%, the splitting tensile strength reached its peak value, which increased by 46%, while peak strain did not change significantly, and the tensile–compressive ratio of specimens keeps rising. It shows that the increase in the polymer–cement ratio had a greater effect on the splitting tensile strength than that of compressive strength.

### 3.4. SEM Analysis

Photographs of polymers with different morphologies in the fiber transition zone are shown in Figure 12. Figure 12b was obtained by enlarging the interface of the transition zone in Figure 12a. The magnification of Figure 12a,b was 2000× and 10,000×, respectively. In Figure 12b, a large number of polymer particles were deposited on the surface of the fibers and the matrix, and a large number of polymer particles filled fractures in the fibers. Because of the large amount of polymer deposition, the deposited particles were closely stacked, and the polymer tended to connect and form membranes.

Figure 12d was obtained by enlarging the interface of transition zone in Figure 12c. The magnification of Figure 12c,d was 5000× and 40,000×, respectively. In Figure 12d, it can clearly be seen that the outer side of the carbon fibers was tightly wrapped by the matrix. The micro-structure of the matrix was composed of the cement slurry matrix and the fibrous polymer. The fibrous polymer acted as a “microfiber” across the cracks and pores in the matrix. 

Figure 12e,f are micro-topographic photos of two different locations. The magnification of Figure 12e,f was 2000× and 10,000×, respectively. Polymer films can be observed at and near the top of bare carbon fibers in Figure 12e. Polymer films were observed at the bonding interface between carbon fibers and the matrix in Figure 12f. Polymer films have excellent tensile and bonding properties, and their interaction has a beneficial effect on the fiber transition zone.

## 4. Discussion

According to the strength test results, adding polymer emulsion powder and carbon fiber is better than adding carbon fiber within the polymer and with a fiber range of reasonable dosage. In other words, the static strength of carbon fiber-reinforced polymer concrete (CFRPC) is better than that of carbon fiber-reinforced concrete. It also proves the rationality of mixing two modifiers (polymer emulsion powder and carbon fiber). The test results were basically consistent with previous research [13,15,21]. This can be explained as follows. The emulsion formed by mixing the polymer emulsion powder with water can assist the dispersion of carbon fibers. The addition of the emulsion has the following functions: Reduction in the thickness of the water film of the carbon fiber–cement interface;Decreasing the difference in the water–cement ratio in the interface layer and the cement matrix;Enhancing cement adhesion to the fiber surface.

Thus, the properties of the carbon fiber–cement interface are greatly improved. The continuous phase is formed at the interface between carbon fiber and cement, and the thickness of the substrate at the contact surface of carbon fiber is increased, which increases the bonding strength between carbon fiber and the substrate. Finally, the concrete under load carbon fiber is not easily pulled out so that the performance of carbon fiber can be fully utilized. In conclusion, when two kinds of modifiers are rationally mixed, the effect of the modifier on the compressive strength of concrete is obviously better than any single modifier; when two kinds of modifiers are unreasonably mixed (the content of carbon fiber and polymer is higher than the critical value), the negative effects of the modifier will be superimposed. For instance, if the carbon fibers and polymers were excessively added, the compressive strength of the specimens is lower than that of the specimens with the single modifier.

When two kinds of modifiers are mixed reasonably, the effect of the carbon fiber and polymer on concrete is more significant than that of the single modifier. This can be explained from two perspectives. On the one hand, the active ingredients contained in the polymer can effectively improve the working performance of concrete, and help fibers disperse well in concrete; thus, the deformability of concrete is improved [22,23]. On the other hand, polymer can effectively improve the bond property interface between fibers and concrete [24,25,26]. Moreover, a good space structure for a fiber–cement slurry matrix polymer membrane is formed, which enhances the matrix microstructure of concrete and weakens the deformation capacity of concrete to some extent [27,28]. However, when the dosage of a modifier is unreasonable, the mechanical properties of the concrete will be greatly reduced, and the deformability of the concrete will be weakened by adding two modifiers. These effects affect the deformation performance of concrete at the same time.

The test results show that the tensile–compressive ratio and the flexural–compressive ratio of concrete increased significantly when the carbon fiber and polymer emulsion powder were added reasonably. It indicates that carbon fiber and polymer emulsion powder have a more significant effect on the tensile strength and flexural strength of concrete than the compressive strength. This can be explained as follows. The modification mechanism of short-cut carbon fibers and polymer emulsion powder for concrete is that high tensile strength materials (carbon fibers and polymer films) are used to cross the cracks and transfer stress to both ends of the cracks [29,30,31]. By limiting the development of concrete cracks [32,33], the strength of the specimens is improved [34], rather than the strengthening of the matrix material. Owing to the reinforcement mode, the tensile strength and flexural strength of concrete increase more significantly than the compressive strength [35,36], which leads to the rapid increase of the tension–compression ratio and the flexural–compression ratio.

## 5. Conclusions

In this paper, compressive, flexural, and splitting tests for CFRPC specimens with different polymer–cement ratios were carried out using an HYY electro-hydraulic servo material test system. The influence of the polymer–cement ratio on the mechanical properties of CFRPC was studied from the aspects of strength and deformation performance, and the modification mechanism of emulsion powder was discussed. The main conclusions are as follows.

With the increase in the polymer–cement ratio, the peak compressive strain of the specimens increased continuously, and the compressive strength of the specimens first increased first then decreased. When the polymer–cement ratio was 12%, the compressive strength of the specimens decreased, compared with specimens without emulsion powder.

With the increase in the polymer–cement ratio, the flexural and splitting properties of the specimens first increased and then decreased. Within the range of raw material proportions in this paper, the optimum polymer–cement ratio was 8%.By adding emulsion powder, the splitting tensile strength of CFRPC had a most significant increase, followed by flexural strength, while the compressive strength increased slightly, even decreasing at a high dosage.Scanning electron microscopy (SEM) results show that the emulsion powder tended to form a film in CFRPC and enhanced the bond strength between the fiber and the matrix. In this way, the tensile crack resistance of the fiber can be fully utilized.

## Figures and Tables

**Figure 1 materials-12-03530-f001:**
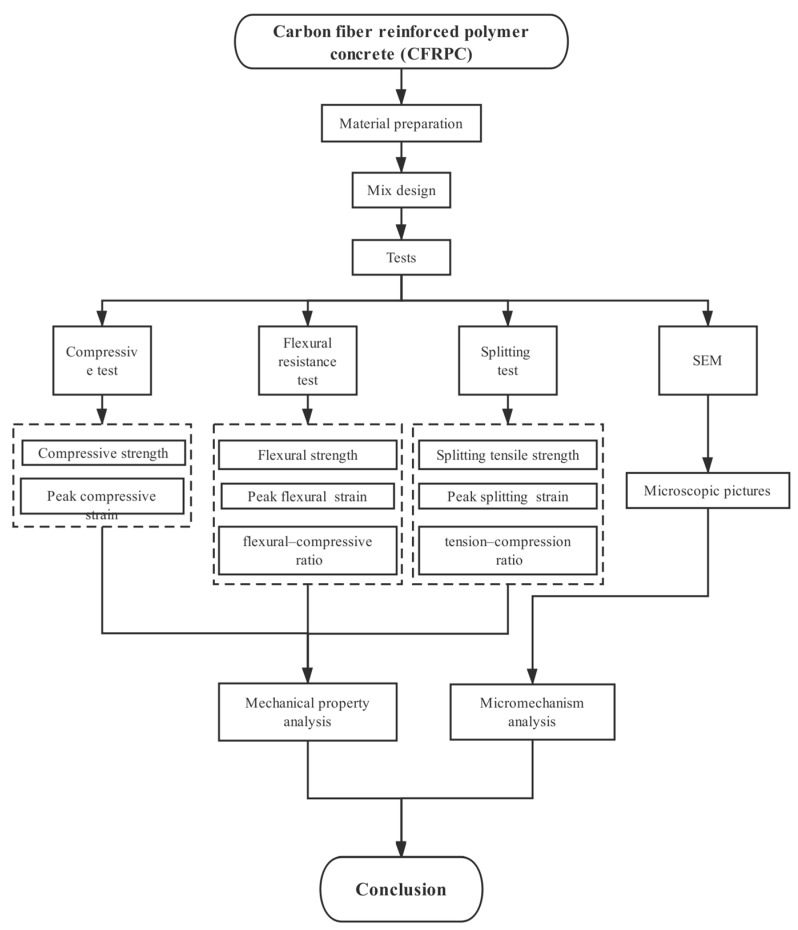
Technical route diagram.

**Figure 2 materials-12-03530-f002:**
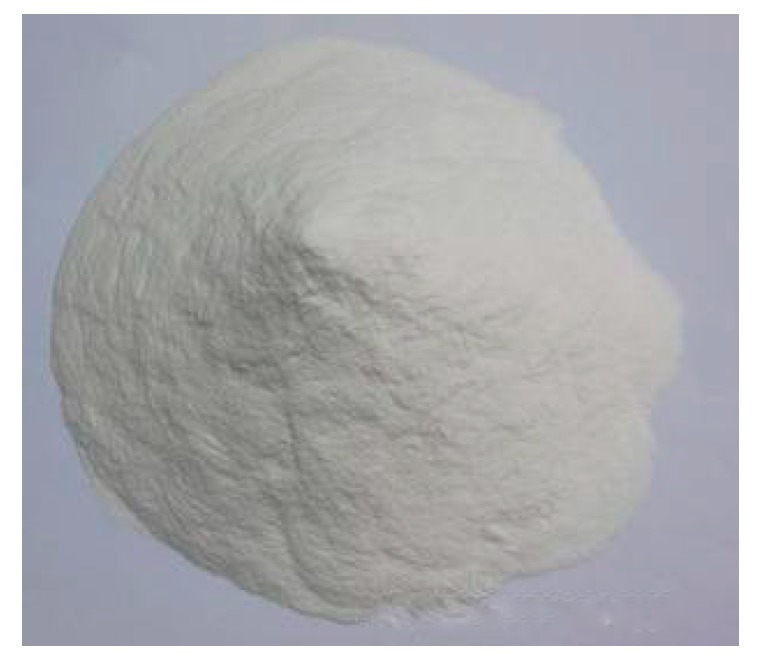
Emulsion powder.

**Figure 3 materials-12-03530-f003:**
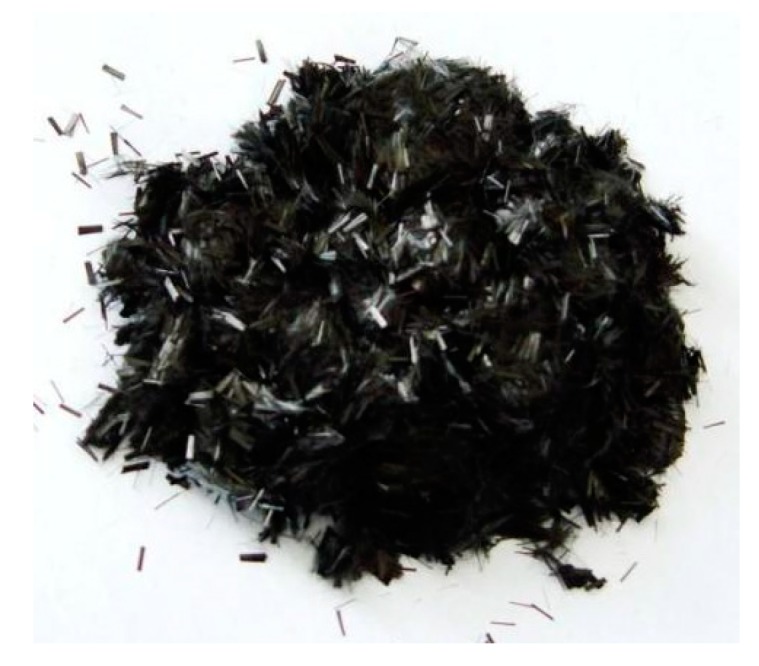
Short-cut carbon fibers.

**Figure 4 materials-12-03530-f004:**
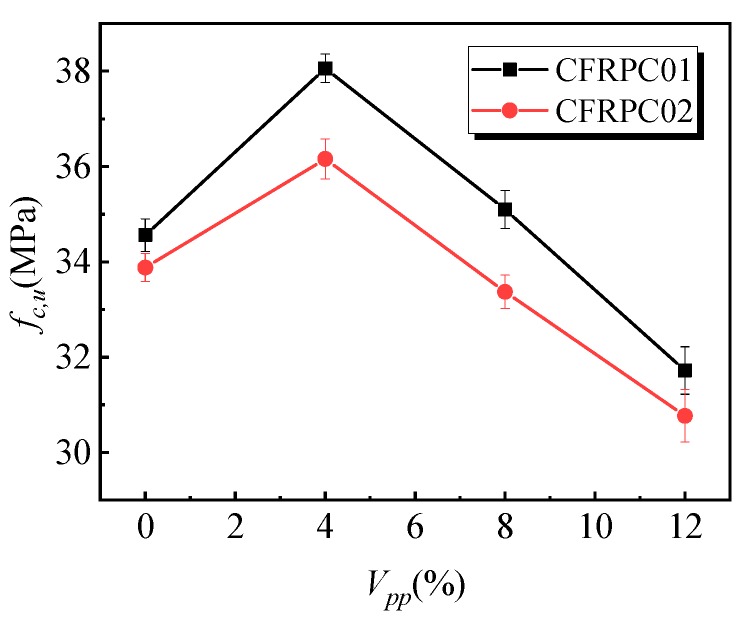
Effect of the polymer–cement ratio on the compressive strength of the specimens.

**Figure 5 materials-12-03530-f005:**
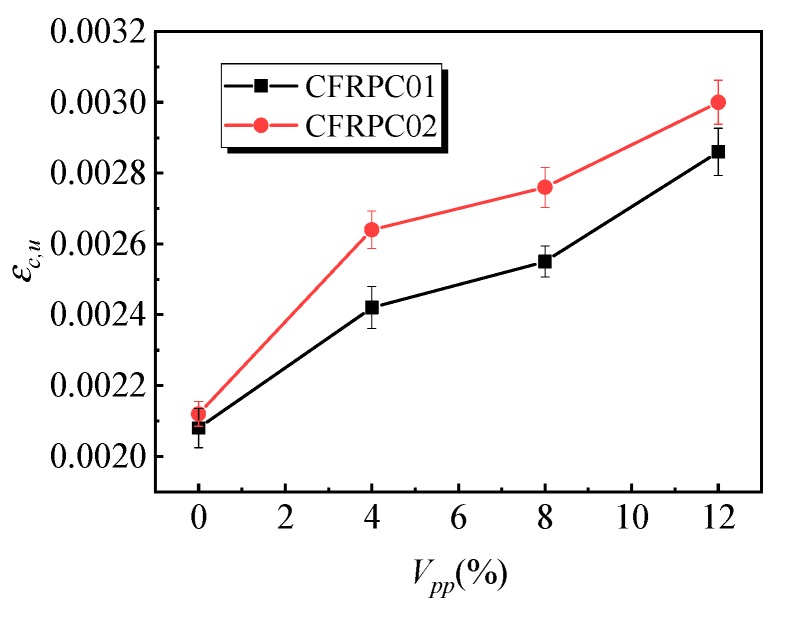
Effect of the polymer–cement ratio on the peak compressive strain of the specimens.

**Figure 6 materials-12-03530-f006:**
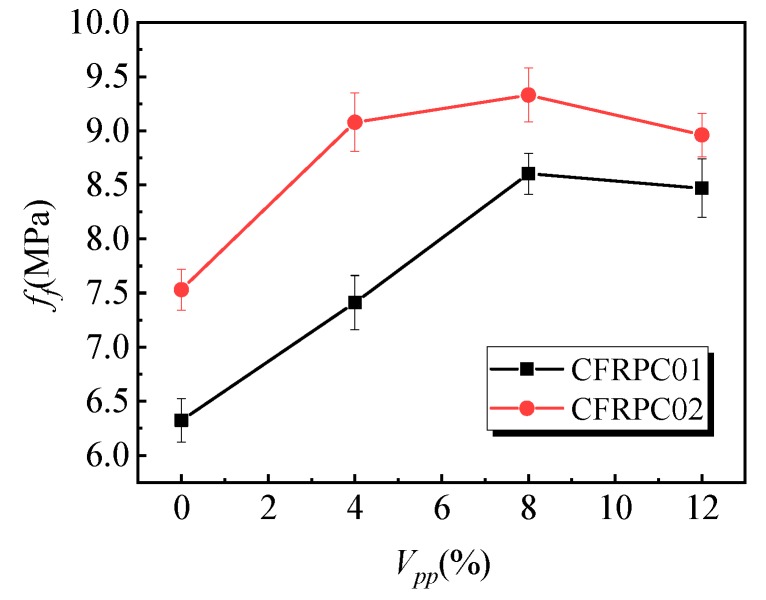
Effect of the polymer–cement ratio on the flexural strength of the specimens.

**Figure 7 materials-12-03530-f007:**
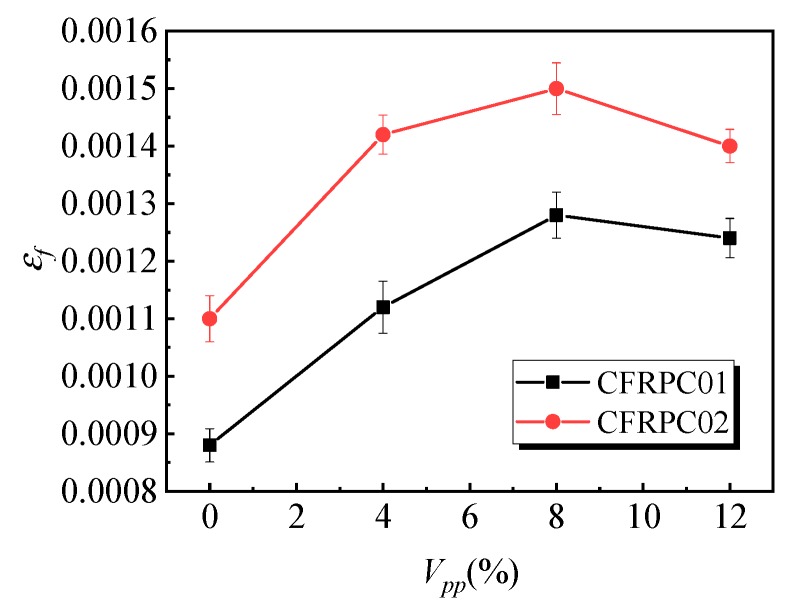
Effect of the polymer–cement ratio on the peak strain in the flexural test.

**Figure 8 materials-12-03530-f008:**
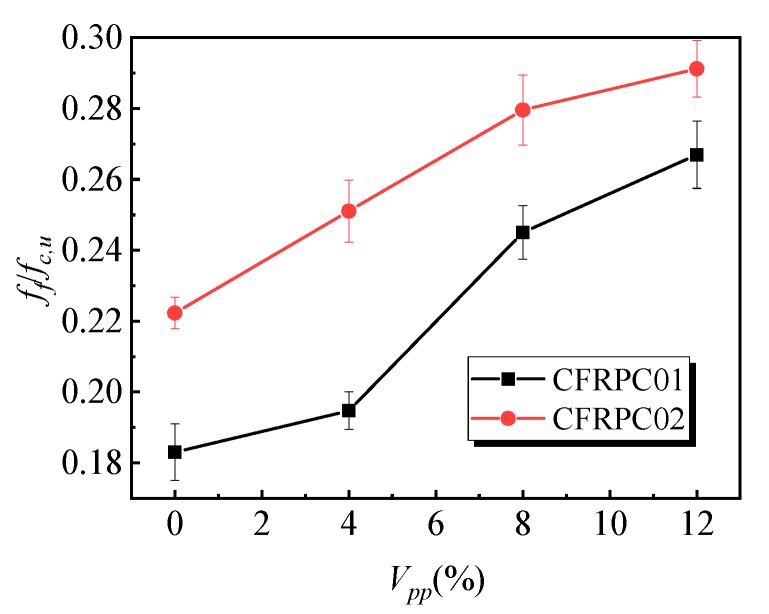
Effect of the polymer–cement ratio on the flexural–compressive ratio of the specimens.

**Figure 9 materials-12-03530-f009:**
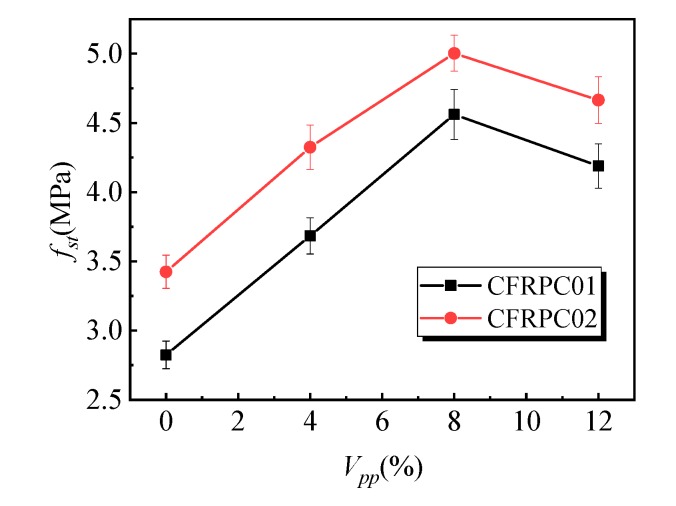
Effect of the polymer–cement ratio on the splitting tensile strength.

**Figure 10 materials-12-03530-f010:**
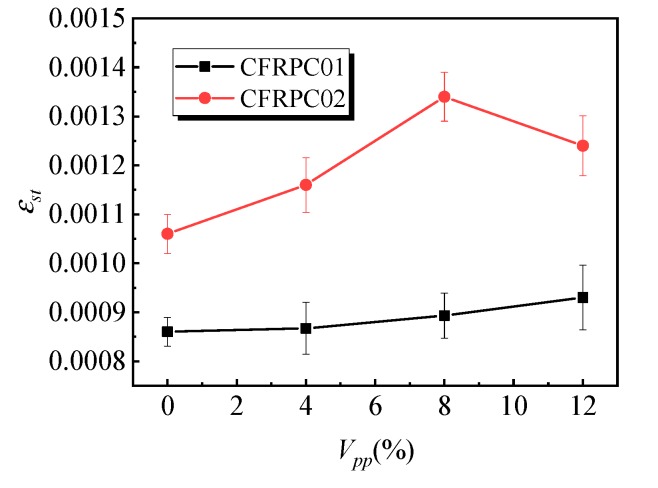
Effect of the polymer–cement ratio on the peak strain of the specimens in the splitting tensile test.

**Figure 11 materials-12-03530-f011:**
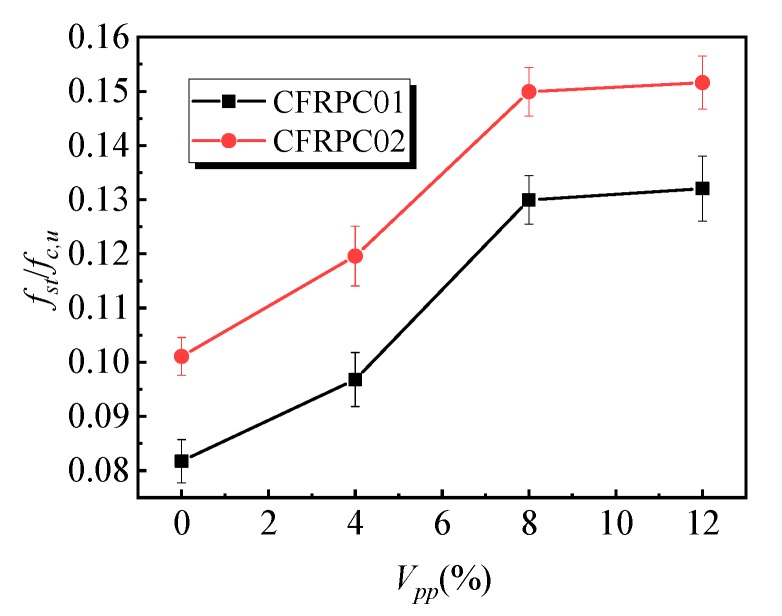
Effect of the polymer–cement ratio on the tension–compression ratio of specimens.

**Figure 12 materials-12-03530-f012:**
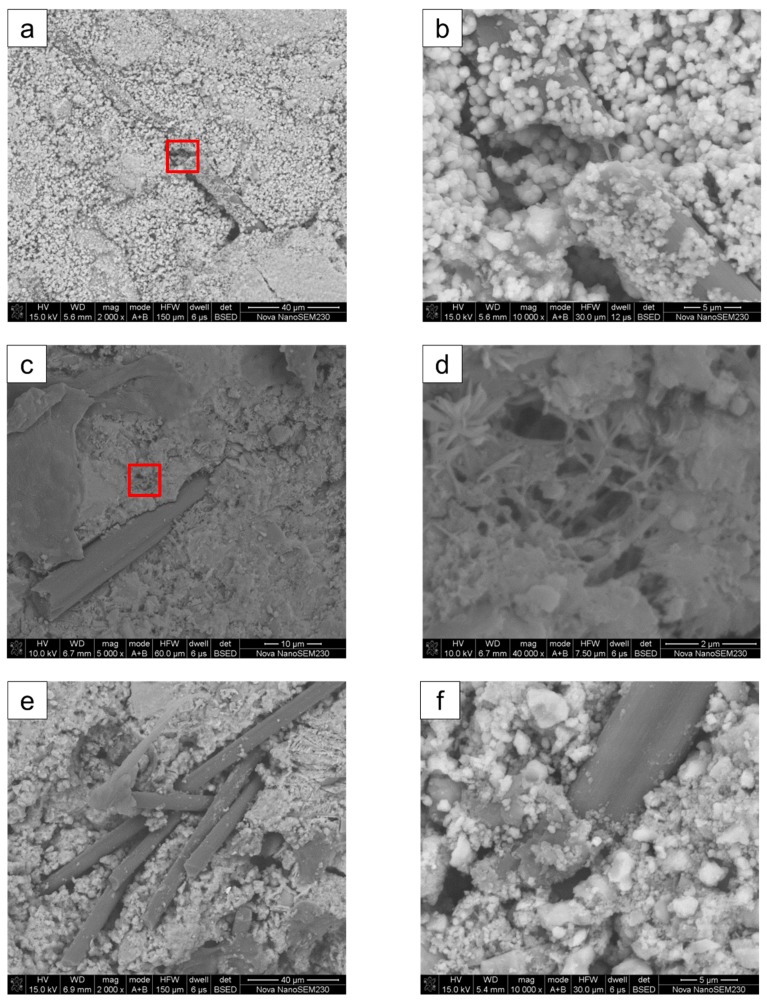
Polymers with different morphologies in the fiber transition zone. (**a**) Granular polymer; (**b**) a 5× enlargement of (**a**); (**c**) fibrous polymer; (**d**) an 8× times enlargement of (**b**); (**e**) membranous polymer on exposed fibers; (**f**) membranous polymer at the fiber–matrix interface.

**Table 1 materials-12-03530-t001:** Main indexes of redispersible emulsion powder.

Appearance	Solid Content	Ash Content	Vitrification Temperature, (°C)	Volume Density, (kg·m^−3^)	Minimum Film Formation Temperature (MFFT), (°C)	Particle Size, (μm)
White powder	≥99%	13 ± 2%	0	400–500	0	1–7

**Table 2 materials-12-03530-t002:** Main indexes of carbon fiber.

Diameter, (μm)	Length, (mm)	Carbon Content, (wt.%)	Elongation at Break, (%)	Tensile Strength, (GPa)	Resistivity, (Ω·cm)	Relative Density, (g·cm^−3^)
7.0 ± 0.2	6	≥93	1.25–1.60	>3.0	1.5 × 10^−3^	1.76

**Table 3 materials-12-03530-t003:** The carbon fiber-reinforced polymer concrete (CFRPC) samples’ compositions.

Test Number	Carbon Fiber	Polymer Emulsion Powder	Cement	Fine Aggregate	Water	Coarse Aggregate	Dispersant	Defoamer	Water-Reducing Agent	Film-Forming Additive
CFRPC01	0.83	0	204	376	100	536	0.82	0.61	2.45	0
		8.17								0.41
		16.33								0.82
		24.50								1.23
CFRPC02	1.66	0								0
		8.17								0.41
		16.33								0.82
		24.50								1.23

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
