# Peer review of "Mechanical Properties of Carbon Fiber-Reinforced Polymer Concrete with Different Polymer–Cement Ratios"

_materials, 2019, doi:10.3390/ma12213530_

Round 1

Reviewer 1 Report

The title of the manuscript is matched to its content.

The structure of the manuscript is proper.

The Introduction sufficiently covers the cases.

In the Reviewer’s opinion, the current state of knowledge relating to the manuscript topic

has been covered and clearly presented.

In the Reviewer’s opinion the manuscript should be published.

Author Response

Thank you for the suggestions. As the reviewer suggested, we revised some of the typos, grammatical errors, vague statements, and long and confusing sentences with the help of an expert to greater fulfil the high standards for publication. More details can be seen in the revised manuscript.

At last, we have also made some other changes in order to improve the manuscript as much as possible. These changes will not influence the content and framework of the paper. And here we did not list the changes but marked them in red in revised manuscript.

Once again, we appreciate for Reviewer’ warm work earnestly, and hope that the revision will meet with approval.

Thank you and best regards.

Sincerely yours,

Gao-Jie Liu

Sep 30, 2019

Reviewer 2 Report

Comment 1 “the cohesion between carbon fiber and cement paste matrix is mainly depended on the crack resistance of fiber” Please explain the logic behind this sentence, usually cohesion/adhesion is dependent on surface energy of the materials not the mechanical properties.

Comment 2 If possible, include chemistry/molecular details of VINNAPAS® 5044N

Comment 3 Please include mechanical test details apart from sample size e.g. head speed of tester, strain rate used etc. For reference, please refer mechanical testing section in paper below and may cite the work : 1) Polymer Volume 148, 18 July 2018, Pages 247-258 https://doi.org/10.1016/j.polymer.2018.06.025

Comment 4: Had authors studied the porosity of sample with and without the emulsifier? If yes, what was the observation, if not please comment what can be the possible outcome based on your study.

Comment 5 Section 3.4 Kindly explain how the samples were obtained for SEM imaging. What type of area was chosen from what type of sample. How was the SEM sample prepared (epoxy embedding or freeze fracture) kindly explain further. This will be useful for readers.

Comment 6 Figure 5 to 12 – Will be nice to have standard deviation for the properties measured in these figures. This is a standard practice.

Comment 7 Suggestion- conclusion in paragraph form instead of bullet point.

Author Response

请参阅附件。

Reviewer 3 Report

The paper is interesting.

The reviewer would recommend the authors to double check the paper in terms of English.

A flowchart would help at the beginning to visualize the process adopted for the research.

Author Response

请参阅附件。

Reviewer 4 Report

The paper concerns the investigations about the mechanical properties of concrete with different additions of carbon fibers and polymer. The study described is very simple from the methodological point of view, only three mechanical properties were tested, and SEM observations were conducted. In general, the main structure of the paper is correct. In the article there are some shortcomings, which should be corrected before acceptance the paper for publication. Detailed comments are listed below:

The style and the language should be improved. There are few syntax and grammar errors. I suggest to check the paper by the English native speaker. Table 1 and 2 - If these are data obtained from the manufacturer and not tested by yourself, please add this information to the text. Table 3 - it seems to me that the table caption is incorrect. Line 102 - how do you know that good dispersion has been achieved? what research methods were used to assess this? Fig 5, 7 and 10 - how many samples are these results? Please provide a basic measure of the dispersion of results, e.g. standard deviation or coefficient of variation. Without this information it is impossible to assess the correctness and repeatability of the results obtained. Line 219-223 - The described effect is caused by any fibers used for the cement matrix, from macrofibers, e.g., steel, polypropylene, to nanofibers, e.g. carbon nanotubes. Authors should add this information to the text, examples of references to literature in this topic:

"Mechanical and durability properties of high-strength concrete containing steel and polypropylene fibers." Construction and building materials 94 (2015): 73-82.

"Evaluation of cracking patterns of cement paste containing polypropylene fibers." Composite Structures 220 (2019): 402-411.

"Recycled glass fiber reinforced polymer additions to Portland cement concrete." Construction and Building Materials 146 (2017): 238-250.

Reviewer 5 Report

Paper may be considered for publication after the following major revisions are considered:

L28: There should be a space between the text and the reference bracket. The paper is hard to understand and there are significant issues with the language. I suggest a review from a native English speaker.  L76-78: Can you explain what these are and what their roles are? Table 1-3: Can you explain or list standards as to how these are determined? L100: Do not instruct. Figure 3, 4: These do not add anything and I would remove. Figure 5 and others: How many specimens were tested? What is the standard deviation? There appears to be no correlation between what is shown in Figure 4 and Figure 5 and what is written in the text. Text says strength is maximum at 4% polymer and is 38.1 MPa, figure shows maximum of 5 MPa at 8% polymer.  Please provide references to literature and compare your findings and insights with those from literature.  The list of references needs to be increased.

Round 2

Reviewer 4 Report

All comments have been addressed. Please make sure the article has been formatted according to the template - e.g. line 118-129 is not. Add Ref. 19 - there is number but there is no bibliographical data. Line 128-129 - please delete the last sentence of the paragraph.

Reviewer 5 Report

Language and presentation need work. The paper is still hard to follow and understand. 

Some of the items in the materials and methods section are not clear - film-forming auxiliaries is used once, but I assume additives is meant. What are all these different materials? Please clearly explain what they are and what they do. 

How were electron microscopy and other tests done? Please clearly describe. These should be in the methods section NOT in the results section. 

L226-227: How? Not clear. Please elaborate. 

I still think the paper is poorly referenced (only 23 references; also note reference 19 is missing) and the level of discussion is minimal (only 5 references in the discussion, similarly very few in the results). 

Author Response

请参阅附件。

Round 3

Reviewer 5 Report

Can be accepted for publication.